# 🐱📖 AGENT KB: LEVERAGING CROSS-DOMAIN EXPERIENCE FOR AGENTIC PROBLEM SOLVING

## ABSTRACT

AI agent frameworks operate in isolation, forcing agents to rediscover solutions and repeat mistakes across different systems. Despite valuable problem-solving experiences accumulated by frameworks like smolagents, OpenHands, and OWL, this knowledge remains trapped within individual systems, preventing collective intelligence emergence. Current memory systems focus on individual agents or framework-specific demonstrations, failing to enable cross-architecture knowledge transfer. We introduce AGENT KB, a universal memory infrastructure enabling seamless experience sharing across heterogeneous agent frameworks without retraining. AGENT KB aggregates trajectories into a structured knowledge base and serves lightweight APIs. At inference time, hybrid retrieval operates through two stages: planning seeds agents with cross-domain workflows, while feedback applies targeted diagnostic fixes. A disagreement gate ensures retrieved knowledge enhances rather than disrupts reasoning, addressing knowledge interference in cross-framework transfer. We validate AGENT KB across major frameworks on GAIA, Humanity's Last Exam, GPQA, and SWE-bench. Results show substantial improvements across diverse model families: compared to baseline pass@1, smolagents with AGENT KB achieve up to 18.7pp gains at pass@3 ($55.2\% \rightarrow 73.9\%$), while OpenHands improves 4.0pp on SWE-bench pass@1 ($24.3\% \rightarrow 28.3\%$). Similar improvements are observed across all base model families. Ablations confirm that hybrid retrieval and feedback stages are essential, with automatically generated experiences matching manual curation. This establishes the foundation for collective agent intelligence through shared memory infrastructures.

## 1 INTRODUCTION

Modern AI agents excel at complex reasoning and tool use (Chan et al., 2023; Hong et al., 2023; Guo et al., 2024; Liu et al., 2025b; Zhou et al., 2023; 2024b), yet each framework operates in isolation, unable to leverage solutions discovered by others. Current memory systems strengthen individual agents (Xu et al., 2025; Packer et al., 2023; Hu & Ying, 2025) or synthesize framework-specific demonstrations (Zheng et al., 2023; Tan et al., 2025), while cross-task approaches like Learn-by-Interact (Su et al., 2025) and A-Mem (Xu et al., 2025) remain confined within single frameworks. This fragmentation forces agents to solve identical problems and make the same mistakes repeatedly.

Enabling cross-framework knowledge sharing requires overcoming three fundamental challenges that no existing system addresses simultaneously. **(1) Representation heterogeneity:** different frameworks organize, encode, and abstract experiences in incompatible ways, which prevents direct transfer or reuse. **(2) Context mismatch:** a solution effective in one tool ecosystem may be invalid or incomplete when transplanted to another, due to differences in available APIs, reasoning protocols, or execution environments. **(3) Knowledge interference:** naively injecting external experiences risks destabilizing the agent's own reasoning flow, producing incoherent plans or compounding errors. Addressing these issues is crucial for building the first interoperable layer of shared memory that enables agents to accumulate and reuse collective intelligence across diverse architectures. Figure 1 contrasts the baseline and AGENT KB-assisted workflows on a representative protein-distance task to ground these challenges.

We introduce AGENT KB, the *first cross-framework plug-and-play knowledge base* that enables seamless experience sharing across heterogeneous agent frameworks without retraining. By distilling

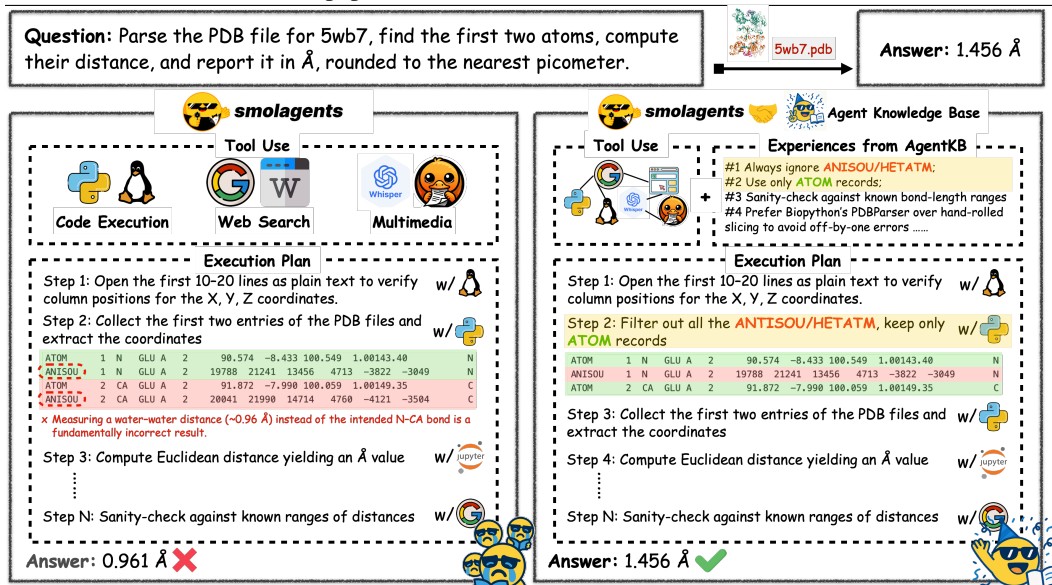

Figure 1: **Agent workflow comparison for PDB distance calculation with and without AGENT KB. (A) Original pipeline:** indiscriminately reads the first two ATOM/HETATM/ANISOU lines, often selecting solvent records and yielding a spurious O–H distance ( 0.961 Å). **(B) AGENT KB-enhanced agent workflow:** applies experience-driven rules—filter out all ANISOU/HETATM, use only genuine ATOM entries in file order, and sanity-check against known N–CA bond-length ranges—to correctly extract the backbone N–CA pair and report the distance of 1.456 Å.

heterogeneous agent trajectories into structured experience units through *framework-agnostic abstraction*, AGENT KB exposes them through lightweight APIs that seamlessly integrate with diverse frameworks, including smolagents (Zhu et al., 2025), OWL (Hu et al., 2025), SWE-Agent (Yang et al., 2024), and OpenHands (Wang et al., 2024a). This creates a continuously growing repository of collective intelligence. To address knowledge interference in cross-framework transfer, we introduce a novel *disagreement gate* mechanism that selectively integrates only coherent updates, ensuring stability and safety during knowledge integration.

**Contributions.** (1) We present the *first cross-framework plug-and-play knowledge base* that integrates with four representative open-source agent frameworks *without requiring retraining or architectural modifications*, demonstrating universal applicability across heterogeneous agent ecosystems. (2) We introduce a novel *disagreement gate mechanism* that addresses the critical challenge of knowledge interference in cross-framework transfer, ensuring stable integration of external experiences. (3) We develop a two-stage retrieval system that enables both planning guidance and feedback-driven refinement while maintaining framework compatibility. (4) We provide comprehensive empirical validation across four distinct agent frameworks on GAIA (Mialon et al., 2023), Humanity's Last Exam (Bio/Chem)[1] (Skarlinski et al., 2025), GPQA (Rein et al., 2024), and SWE-bench (Jimenez et al., 2023), establishing the first systematic study of cross-framework knowledge transfer effectiveness.

Evaluation across reasoning and software engineering tasks demonstrates that AGENT KB consistently boosts diverse agent–model combinations. On GAIA, smolagents improve by up to 18.7pp at `pass@3` (55.2% → 73.9%), while on SWE-bench Lite, OpenHands gains 4.0pp at `pass@1` (24.3% → 28.3%). On HLE, OpenHands outperforms specialized systems (9.5% → 14.1% at `pass@3`) and on GPQA, it improves `GPT-4.1` from 62.6% to 72.7%. Similar trends appear for both open-source models (`Qwen`, `DeepSeek`) and proprietary backbones (`GPT`, `Claude`), underscoring the broad applicability of our approach. Ablations further show that automatically distilled experiences perform on par with hand-curated ones, confirming AGENT KB as a scalable path toward collective agent intelligence.

## 2 RELATED WORK

### 2.1 AGENTIC MEMORY SYSTEMS

Memory systems have progressed from simple storage to advanced architectures that support complex reasoning (Piao et al., 2025; Zeng et al., 2024; Liu et al., 2025b; Du et al., 2025; Wu et al., 2025b; Zhang et al., 2025), though current systems still struggle with managing large amounts of information and transferring knowledge (Wang et al., 2024c). Earlier systems embedded knowledge in a latent space (Wang et al., 2024b), while newer, more organized approaches have adopted graph-based and hierarchical frameworks (Xu et al., 2025; Anokhin et al., 2024; Packer et al., 2023; Hu & Ying, 2025). Improving retrieval beyond basic RAG (Lewis et al., 2020a;b) includes various innovations for

---

[1] https://huggingface.co/datasets/futurehouse/hle-gold-bio-chem

Figure 2: **End-to-end workflow of AGENT KB.** (a) **Construction**: heterogeneous agent trajectories and few-shot human seeds are abstracted into structured experiences and indexed in the AGENT KB. (b) **Evolution**: AGENT KB expands across domains through addition, conflict resolution, and timely eviction, maintaining quality while scaling. (c) **Solving tasks**: agents apply a two-stage `Reason-Retrieve-Refine` loop, planning with retrieved workflows and refining via feedback.

indexing, temporal cues, semantic tagging, and chunking strategies (Huang et al., 2025b; Gutiérrez et al., 2024; Liu et al., 2025a; Salama et al., 2025; Hu et al., 2024). These methods have also evolved to incorporate neuroscience inspiration or multi-agent variants (Ye, 2025; Wang et al., 2025; Squire et al., 2015; Wang et al., 2024d; Zhu et al., 2024; Qiao et al., 2024; Xu et al., 2024; Chen et al., 2025; Ganguli et al., 2025; Lv et al., 2024; Shuster et al., 2021; Niu et al., 2024; Mala et al., 2025). Despite improvements, existing approaches face significant limitations: they are primarily tailored for single agents, retain separate memory systems that prevent shared knowledge, and lack ways to reuse experiences across different areas, which makes them vulnerable in new or unfamiliar contexts.

## 2.2 AGENTIC KNOWLEDGE TRANSFER

Alongside memory infrastructures, another research area condenses agent trajectories into workflow priors that guide future problem solving: retrieval-based systems (Zheng et al., 2023; Zhou et al., 2024a) stabilize tool use with exemplar traces, mined sub-workflows support reuse across tasks (Wang et al., 2024d), and templating pipelines refine plans within narrow families (Tan et al., 2025; Liu et al., 2025e). Knowledge-augmented and collaborative planners extend these ideas with structured repositories (Zhu et al., 2024; Qiao et al., 2024; Liu et al., 2025c), yet efforts for multi-agent memory still maintain siloed stores even when coordination leverages in-context learning or RAG (Lu et al., 2023; Zhong et al., 2024; Glocker et al., 2025). Early evidence shows that cross-agent transfer depends on experience quality and stronger-to-weaker sharing (Shah et al., 2025; Zhao et al., 2025; Alakuijala et al., 2025), while visions for lifelong cognition call for AI-native, adaptive, evaluable, case-based infrastructures that jointly encode problem patterns, workflows, metadata, and relational structure (Wang et al., 2024c; Wei et al., 2025b; Pink et al., 2025; Hatalis et al., 2025). AGENT KB abstracts heterogeneous trajectories into hierarchical experience units and couples them with hybrid retrieval, seeding planning with cross-domain workflows, and injecting feedback corrections to enable transfer across divergent tasks. Although recent reviews of case-based reasoning for LLM agents (Hatalis et al., 2025) emphasize the classic `Retrieve-Reuse-Revise-Retain` cycle, our framework restructures it as a `Reason-Retrieve-Refine` loop with write-back across agents. In parallel, systems such as DSPy (Khattab et al., 2023) offer prompt declarative programming for task workflows; AGENT KB is complementary, providing a reusable cross-framework memory layer that can be integrated beneath DSPy or similar pipelines to capture and transfer execution knowledge.

## 3 METHODOLOGY

### 3.1 OVERVIEW

AGENT KB enables agents to learn from collective experiences across tasks and frameworks by capturing execution traces and abstracting them into reusable experiences. When facing new tasks, agents retrieve relevant past experiences to guide the refinement of planning and execution, transforming individual interactions into shared, cumulative intelligence. Our approach operates through two key stages: initial solution planning, which utilizes past experiences, and feedback-driven execution

improvements, both employing the same `Reason-Retrieve-Refine` structure but with different query formulations: the first targeting task descriptions and the other execution feedback patterns.

To ensure framework-agnosticism, AGENT KB standardizes the experience schema and provides lightweight REST endpoints to submit and retrieve experiences. This enables heterogeneous agents to contribute and consume the same knowledge base without requiring architectural changes, allowing the sharing and reuse of high-quality entries that collectively build intelligence across the ecosystem.

## 3.2 SELF-EVOLVING AGENT KB

**Experience Representation.** We transform agent execution logs into structured experiences through human-guided abstraction:

$$E = \langle \pi, \gamma, S, \mathcal{C} \rangle \tag{1}$$

where $\pi$ is the task embedding via `all-MiniLM-L6-v2` from `sentence-transformers`, $\gamma$ encodes goal constraints as structured predicates, $S = \{(a_i, r_i)\}$ stores action–reasoning pairs, and $\mathcal{C}$ carries metadata for cross-framework compatibility. Abstraction is done with few-shot prompting (10–15 human-curated exemplars per domain) and standardized action vocabularies across frameworks (e.g., smolagents (Zhu et al., 2025), OWL (Hu et al., 2025), SWE-Agent (Yang et al., 2024), OpenHands (Wang et al., 2024a)). Full construction details appear in Appendix B, while the prompt templates are compiled in Appendix F with general generation patterns in Appendix F.1 and pipeline variants in Appendix F.2.

**Self-Evolving Memory.** The memory evolves through addition, deduplication, and eviction (Fig. 2b). AGENT KB grows as diverse agent frameworks contribute execution experiences, creating cumulative intelligence that expands both coverage and generalization capability between domains. When a candidate is highly similar to existing entries $\max_{\pi' \in \mathcal{E}} \cos(\pi, \pi') > \tau$ (default $\tau = 0.8$), an LLM ranker compares reasoning quality, completeness, and transferability to keep the superior entry, preventing redundancy. Under memory pressure, experiences follow an adaptive eviction policy with a learned utility score that balances recency, frequency, and cross-framework transferability. Each experience $E_j$ maintains $u_j \leftarrow u_j + \eta(r_j - u_j)$, where $r_j$ is the reward signal (e.g., retrieval success or execution gain) and $\eta$ is a learning rate. Low-utility entries are evicted, yielding dynamic memory allocation that preserves high-value cross-domain knowledge.

## 3.3 EXECUTION VIA `Reason-Retrieve-Refine`

As shown in Fig. 2c, AGENT KB intercepts reasoning at *planning* and *feedback* stages while leaving the base agent unchanged. Retrieved experiences are adapted through the `Reason-Retrieve-Refine` cycle.

**Retrieval Pipeline.** Retrieval uses two complementary filters: (1) lexical retrieval (BM25) to shortlist candidates with domain/tool compatibility, and (2) semantic ranking by task similarity via `all-MiniLM-L6-v2` embeddings. We also support a calibrated hybrid fusion (default $\alpha = 0.5$) of the two scores:

$$\sigma_i^{\text{hyb}} \leftarrow \alpha \cdot \tilde{\sigma}_i^{\text{text}} + (1 - \alpha) \cdot \tilde{\sigma}_i^{\text{sem}}, \quad \alpha \in [0, 1].$$

After reranking, the top-$k$ candidates are deduplicated before refinement.

**Planning Stage.** The system applies the `Reason-Retrieve-Refine` cycle to the incoming task description to generate a preliminary execution plan. In the `Reason` step, it surfaces key requirements and potential challenges, producing structured queries that target reusable subroutines or successful completion patterns. The `Retrieve` step then selects past experiences via the hybrid similarity scorer, providing candidate trajectories. Rather than direct reuse, these candidates are adapted in the `Refine` step: abstract action patterns are aligned with currently available tools and APIs using metadata $\mathcal{C}$ for cross-framework compatibility. When multiple experiences are retrieved, the system synthesizes them into a coherent workflow by applying *entity mapping*, *tool substitution*, and *step reordering* to satisfy domain-specific constraints—enabling cross-framework compatibility without sacrificing execution fidelity. The final output is an executable plan $\rho$, which is returned to the agent framework for execution in its native environment.

**Feedback Stage** After initial execution, AGENT KB re-engages through a second `Reason-Retrieve-Refine` cycle. In the `Reason` step, it analyzes execution traces to identify errors, bottlenecks, or unexpected outcomes. The `Retrieve` step then produces queries derived from these traces, rather than

task descriptions, that favor experiences that document successful refinements in similar contexts. In the `Refine` step, candidate fixes are adapted to live execution, taking into account domain constraints and observed errors. To ensure safety, we introduce a *disagreement gate*:

$$\mathcal{G}(\rho, \rho') = \mathbb{1}\big[\cos\big(\phi(\rho), \phi(\rho')\big) \geq \beta\big],$$

where $\rho$ is the original plan, $\rho'$ the refined one, $\phi$ is implemented as `all-MiniLM-L6-v2` embeddings, and $\beta = 0.8$ by default. Only refinements with $\mathcal{G}(\rho, \rho') = 1$ are applied.

## 4 EXPERIMENT

### 4.1 SETUP

**Dataset** We benchmark AGENT KB on four suites covering reasoning and software engineering. GAIA (Mialon et al., 2023) contributes 165 tasks: 53 Level 1 (factual lookup), 86 Level 2 (multi-step reasoning), and 26 Level 3 (analysis and synthesis). The cleaned biology & chemistry subset provides 149 tasks needing multimodal reasoning with scientific images and tools. GPQA (Rein et al., 2024) offers 198 graduate-level MCQs in physics, chemistry, and biology, requiring experts. Both GAIA and HLE support web browsing, file I/O, and tools within standard limits, whereas GPQA focuses on reasoning without the use of external tools. Benchmarks are reported with `pass@1`, `pass@2`, and `pass@3` accuracy. SWE-bench Lite (Jimenez et al., 2023) comprises 300 GitHub issues across Python repositories; success is tested with 50- and 100-iteration limits for reproducibility. It restricts network access and limits operations to those within the repository. These benchmarks show how knowledge refinement affects reasoning (GAIA, HLE, GPQA) and software engineering (SWE-bench).

**AGENT KB Construction.** We bootstrap AGENT KB with 80 seed trajectories written by five computer-science graduate students (60 BrowseComp/HopRAG-style browsing traces and 20 SWE-Gym-style coding logs). These curated demonstrations are never retrieved directly; instead they guide automatic rollouts executed by smolagents (Zhu et al., 2025), OWL (Hu et al., 2025), SWE-Agent (Yang et al., 2024), and OpenHands (Wang et al., 2024a) across BrowseComp (Wei et al., 2025a), HopRAG (Liu et al., 2025d), HLE[2] (Phan et al., 2025), WebWalkerQA (Wu et al., 2025a), RepoClassBench (Deshpande et al., 2024), SWE-Gym-Raw (Pan et al., 2024), and RepoEval (Zhang et al., 2023). We normalize both successful and failed runs into structured experience units, yielding roughly 9k workflow summaries and 7k execution snippets before evaluation. This mix equips the memory with reusable plans for each modality and the diagnostic traces in later passes (Appendix A).

**Model Configurations** We attach the same AGENT KB instance to all planners through lightweight RPC calls so that experiences gathered in one framework are *instantly available to the others*, demonstrating true cross-framework knowledge transfer. GAIA experiments use smolagents (backed by `GPT-4o`, `GPT-4.1`, `Claude-3.7`, `Qwen-3 32B`, and `DeepSeek-R1`) and OWL (`GPT-4o`); SWE-bench Lite is evaluated with SWE-Agent (`GPT-4.1`, `o3-mini`) and OpenHands (`GPT-4o`, `GPT-4.1`, `Claude-3.7`, `Qwen-3 32B`, `DeepSeek-R1`, `o3-mini`). Each benchmark instance is solved in three sequential passes: `pass@1` retrieves cross-task experiences without exposure to held-out labels, `pass@2` enriches AGENT KB with failure diagnoses from the first attempt, and `pass@3` revisits unresolved cases using the expanded retrieval pool. Unless otherwise stated, we fix the base model to `GPT-4.1`, the temperature to 1.0, and the retrieval top-$k$ to 3, mirroring the setting used for Figure 3. We estimate budget caps assuming OpenAI pricing ($1.36/M prompt, $5.44/M completion tokens); see Appendix D for detailed cost analysis.

### 4.2 MAIN RESULTS

Table 1 shows that AGENT KB delivers consistent gains across *heterogeneous* agent stacks and model families on GAIA. With smolagents, `GPT-4.1` rises from 55.2% to 73.9% `pass@3` accuracy (+18.7), with the largest lift at Level 2 (53.5% → 73.3%, +19.8). The more capable `Claude-3.7` backbone reaches 75.2% `pass@3` and adds 19.2 on Level 3 (38.5% → 57.7%), matching or exceeding closed-source systems such as h2oGPTe (63.6%). Relative to A-Mem (Xu et al., 2025), which lifts `GPT-4o` smolagents to 69.1%, AGENT KB attains 73.9% with the same planner, indicating hybrid retrieval extracts more value from each pass. OWL with `GPT-4o` also benefits: accuracy improves by 20.0 overall (43.6% → 63.6%) and retains gains on the most challenging questions (30.8% → 38.5%).

On SWE-bench Lite, Table 2b highlights similar trends. `GPT-4.1` paired with SWE-Agent improves from 24.3% to 38.0% at 50 iterations and 42.3% under 100 iterations. OpenHands sees

---

[2]We deliberately removed the biology & chemistry subset from Humanity's Last Exam as a test set.

Table 1: Results on GAIA benchmark (val set). We report pass@1 for all standard baselines. For methods that build on top of a base framework (A-MEM (Xu et al., 2025) and AGENT KB), we present the baseline alongside the improvements achieved by each enhanced variant.

| Method | Models | Config | Average | Level 1 | Level 2 | Level 3 |
|---|---|---|---|---|---|---|
| ***Agentic Model*** | | | | | | |
| Search-o1-32B (Li et al., 2025a) | Qwen-3 | pass@1 | 39.8 | 53.8 | 34.6 | 16.7 |
| WebThinker-32B-RL (Li et al., 2025b) | Qwen-3 | pass@1 | 48.5 | 56.4 | 50.0 | 16.7 |
| ***Closed-source Frameworks*** | | | | | | |
| TraseAgent (Trase, 2024) | Claude-3.5 | pass@1 | 70.3 | 83.0 | 69.8 | 46.2 |
| Deep Research (OpenAI, 2024) | Unknown | pass@1 | 67.4 | 74.3 | 69.1 | 47.6 |
| h2oGPTe (H2O.ai, 2024) | Claude-3.5 | pass@1 | 63.6 | 67.9 | 67.4 | 42.3 |
| Desearch (AI, 2024) | GPT-4o | pass@1 | 57.0 | 71.7 | 58.1 | 23.1 |
| Alita (Qiu et al., 2025) | Claude-3.7 | pass@1 | 72.7 | 81.1 | 75.6 | 46.2 |
| ***Open-source Frameworks*** | | | | | | |
| OWL (Hu et al., 2025) | o3-mini | pass@1 | 60.6 | 81.1 | 58.1 | 26.9 |
| TapeAgents (Bahdanau et al., 2024) | Claude-3.7 | pass@1 | 55.8 | 71.7 | 53.5 | 30.8 |
| AutoAgent (Tang et al., 2025) | Claude-3.5 | pass@1 | 55.2 | 71.7 | 53.4 | 26.9 |
| Magnetic-1 (Fourney et al., 2024) | o1 | pass@1 | 46.1 | 56.6 | 46.5 | 23.1 |
| FRIDAY (Wu et al., 2024b) | GPT-4 turbo | pass@1 | 34.6 | 45.3 | 34.9 | 11.5 |
| smolagents (Roucher et al., 2025) | GPT-4o | pass@1 | 43.6 → 57.0 ↑13.4 | 52.8 → 71.7 ↑18.9 | 41.9 → 57.0 ↑15.1 | 30.8 → 26.9 ↓3.9 |
| ↝ +A-MEM (Xu et al., 2025) | | pass@2 | 53.9 → 64.2 ↑10.3 | 64.2 → 83.0 ↑18.9 | 53.5 → 64.0 ↑10.5 | 34.6 → 26.9 ↓7.7 |
| | | pass@3 | 57.0 → 69.1 ↑12.1 | 69.8 → 86.8 ↑17.0 | 55.8 → 69.8 ↑14.0 | 34.6 → 30.8 ↓3.8 |
| smolagents (Roucher et al., 2025) | GPT-4.1 | pass@1 | 55.2 → 61.2 ↑6.1 | 67.9 → 79.3 ↑11.3 | 53.5 → 58.1 ↑4.7 | 34.6 → 34.6 ↑0.0 |
| ↝ +AGENT KB | | pass@2 | 61.8 → 67.3 ↑5.5 | 73.6 → 83.0 ↑9.4 | 62.8 → 67.4 ↑4.7 | 34.6 → 34.6 ↑0.0 |
| | | pass@3 | 68.5 → 73.9 ↑5.5 | 77.4 → 84.9 ↑7.6 | 68.6 → 73.3 ↑4.7 | 50.0 → 53.9 ↑3.9 |
| smolagents (Roucher et al., 2025) | Claude-3.7 | pass@1 | 58.8 → 65.5 ↑6.7 | 64.2 → 75.5 ↑11.3 | 61.6 → 66.3 ↑4.7 | 38.5 → 38.5 ↑0.0 |
| ↝ +AGENT KB | | pass@2 | 63.6 → 69.7 ↑6.1 | 77.4 → 79.3 ↑1.9 | 61.6 → 69.8 ↑8.1 | 42.3 → 50.0 ↑7.7 |
| | | pass@3 | 72.7 → 75.2 ↑2.4 | 81.1 → 84.9 ↑3.8 | 74.4 → 74.4 ↑0.0 | 50.0 → 57.7 ↑7.7 |
| OWL (Hu et al., 2025) | GPT-4o | pass@1 | 43.6 → 52.7 ↑9.1 | 52.8 → 64.2 ↑11.3 | 41.9 → 54.7 ↑12.8 | 30.8 → 23.1 ↓7.7 |
| ↝ +AGENT KB | | pass@2 | 53.9 → 60.6 ↑6.7 | 64.2 → 75.5 ↑11.3 | 53.5 → 61.6 ↑8.1 | 34.6 → 26.9 ↓7.7 |
| | | pass@3 | 57.0 → 63.6 ↑6.7 | 69.8 → 79.3 ↑9.4 | 55.8 → 61.6 ↑5.8 | 34.6 → 38.5 ↑3.8 |

Table 2: **Results on multiple benchmarks.** We report baseline pass@1 and AGENT KB-enhanced variants.

(a) SWE-bench Lite (Jimenez et al., 2023) (300 instances) with max iterations of 50 and 100.

| Method | | Models | Success Rate (%) Max Iter 50 | Max Iter 100 | Budget Cap |
|---|---|---|---|---|---|
| SWE-agent (Yang et al., 2024) | | | 24.3 | 27.0 | $3.0 |
| pass@1 | +AGENT KB | GPT-4.1 | 31.7 ↑7.4 | 35.3 ↑8.3 | $3.0 |
| pass@2 | +AGENT KB | | 36.7 ↑12.4 | 38.0 ↑11.0 | $3.0 |
| pass@3 | +AGENT KB | | **38.0** ↑13.7 | **42.3** ↑15.3 | $3.0 |
| OpenHands (Wang et al., 2024a) | | | 24.3 | 28.7 | $4.5 |
| pass@1 | +AGENT KB | GPT-4.1 | 28.3 ↑4.0 | 31.7 ↑3.0 | $4.5 |
| pass@2 | +AGENT KB | | 37.3 ↑13.0 | 42.3 ↑13.7 | $4.5 |
| pass@3 | +AGENT KB | | **38.7** ↑14.3 | **45.7** ↑17.0 | $4.5 |
| OpenHands | | | 30.0 | 41.3 | $4.5 |
| pass@1 | +AGENT KB | Claude-3.7 | 46.7 ↑16.7 | 48.3 ↑7.0 | $4.5 |
| pass@2 | +AGENT KB | | 49.7 ↑19.7 | 51.7 ↑10.3 | $4.5 |
| pass@3 | +AGENT KB | | **51.0** ↑21.0 | **53.3** ↑12.0 | $4.5 |

(b) GAIA (Mialon et al., 2023) (165 instances) with pass@1 baseline and AGENT KB-enhanced variants.

| Method | | Models | Accuracy (%) Avg | L1 | L2 | L3 |
|---|---|---|---|---|---|---|
| OWL (Hu et al., 2025) | | | 43.6 | 52.8 | 41.9 | 30.8 |
| pass@1 | +AGENT KB | GPT-4o | 52.7 ↑9.1 | 64.2 ↑11.3 | 54.7 ↑12.8 | 23.1 ↓7.7 |
| pass@2 | +AGENT KB | | 60.6 ↑6.7 | 75.5 ↑11.3 | 61.6 ↑8.1 | 26.9 ↓7.7 |
| pass@3 | +AGENT KB | | **63.6** ↑6.7 | **79.3** ↑9.4 | **61.6** ↑5.8 | **38.5** ↑3.8 |
| smolagents (Roucher et al., 2025) | | | 55.2 | 67.9 | 53.5 | 34.6 |
| pass@1 | +AGENT KB | GPT-4.1 | 61.2 ↑6.1 | 79.3 ↑11.3 | 58.1 ↑4.7 | 34.6 ↑0.0 |
| pass@2 | +AGENT KB | | 67.3 ↑5.5 | 83.0 ↑9.4 | 67.4 ↑4.7 | 34.6 ↑0.0 |
| pass@3 | +AGENT KB | | **73.9** ↑5.5 | **84.9** ↑7.6 | **73.3** ↑4.7 | **53.9** ↑3.9 |
| smolagents | | | 58.8 | 64.2 | 61.6 | 38.5 |
| pass@1 | +AGENT KB | Claude-3.7 | 65.5 ↑6.7 | 75.5 ↑11.3 | 66.3 ↑4.7 | 38.5 ↑0.0 |
| pass@2 | +AGENT KB | | 69.7 ↑6.1 | 79.3 ↑1.9 | 69.8 ↑8.1 | 50.0 ↑7.7 |
| pass@3 | +AGENT KB | | **75.2** ↑2.4 | **84.9** ↑3.8 | **74.4** ↑0.0 | **57.7** ↑7.7 |

(c) Humanity's Last Exam (Bio/Chem) (Skarlinski et al., 2025) (149 instances) with pass@1 baseline and AGENT KB-enhanced variants.

| Method | | Models | Accuracy (%) |
|---|---|---|---|
| AutoGen (Wu et al., 2024a) | | GPT-4.1 | 7.4 |
| SciMaster (Chai et al., 2025) | | GPT-4.1 | 9.5 |
| Biomni (Huang et al., 2025a) | | GPT-4.1 | 10.7 |
| OpenHands (Wang et al., 2024a) | | | 9.5 |
| pass@1 | +AGENT KB | | 10.1 ↑0.7 |
| pass@2 | +AGENT KB | GPT-4.1 | 12.1 ↑2.7 |
| pass@3 | +AGENT KB | | 14.1 ↑4.7 |

(d) GPQA benchmark (Rein et al., 2024) (198 instances) with pass@1 baseline and AGENT KB-enhanced variants.

| Method | | Models | Accuracy (%) |
|---|---|---|---|
| Direct Reasoning | | o3-mini | 75.0 |
| | | Claude-3.7 | 67.4 |
| | | GPT-4.1 | 64.6 |
| OpenHands (Wang et al., 2024a) | | | 62.6 |
| pass@1 | +AGENT KB | | 67.2 ↑4.6 |
| pass@2 | +AGENT KB | GPT-4.1 | 70.7 ↑8.1 |
| pass@3 | +AGENT KB | | **72.7** ↑10.1 |

a 14.4-point increase at 50 iterations (24.3% → 38.7%) and a 17.0-point gain at 100 iterations (28.7% → 45.7%). The strongest backbone, Claude-3.7, achieves the largest jump, adding 21.0 at 50 iterations (30.0% → 51.0%) and 12.0 at 100 iterations (41.3% → 53.3%).

Beyond GAIA and SWE-bench, AGENT KB also improves on challenging scientific QA datasets. On HLE (Table 2c, OpenHands baseline (9.5%) lags behind Biomni (10.7%), but surpasses it once retrieval is applied (12.1% at pass@2, 14.1% at pass@3). On GPQA (Table 2d), OpenHands with GPT-4.1 climbs from 62.6% to 72.7%, approaching latest proprietary models. These improvements are achieved without additional fine-tuning or tool customization, underscoring the *zero-shot transferability* of the shared experience store across diverse agent architectures.

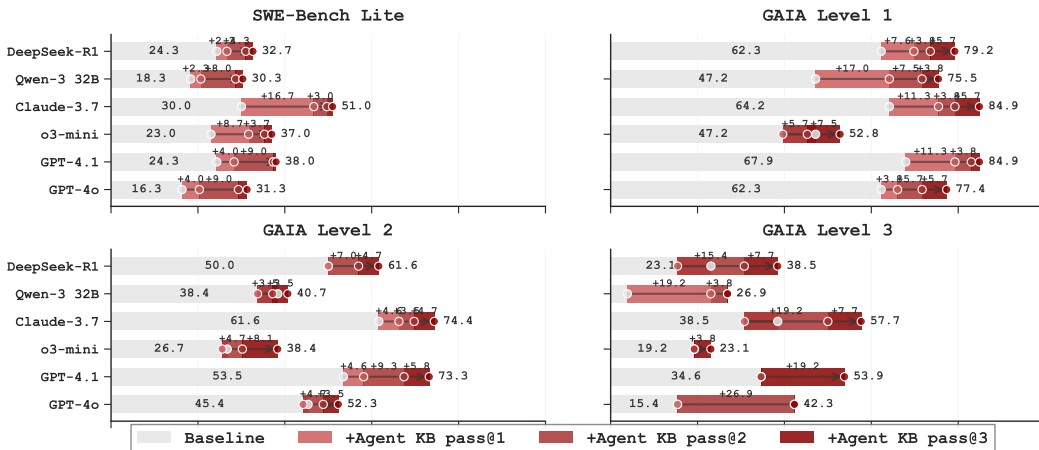

Figure 3: Score improvements (%) across benchmarks for multiple LLMs enhanced with AGENT KB. Results on SWE-Bench Lite (Jimenez et al., 2023) using OpenHands (Wang et al., 2024a) (left) and GAIA benchmark (Mialon et al., 2023) using smolagents (Zhu et al., 2025) (right) showing iterative improvement through progressive knowledge refinement. Red intensity indicates the refinement stage, baseline performance in gray.

The stacked analysis in Figure 3 confirms that every backbone benefits from successive retrieval passes across reasoning (GAIA, HLE, GPQA) and software engineering (SWE-bench). For example, GPT-4o gains 15.0 on SWE-bench (16.3% → 31.3%) and 26.9 on GAIA Level 3 (15.4% → 42.3%). GPT-4.1 delivers similarly strong lifts on GAIA Level 2 (53.5% → 73.3%), while Claude-3.7 records the largest SWE-bench improvement (30.0% → 51.0%). Across all settings, pass@1 supplies the initial boost by importing compatible workflows, while deeper passes (pass@2/pass@3) contribute targeted refinements, most pronounced on scientific and general reasoning tasks.

Representative GAIA and SWE-bench walkthroughs illustrating these dynamics appear in Appendix E, with the full execution trace reproduced in Appendix E.1.

### 4.3 ABLATION STUDIES

**Impact of Retrieval Passes and Reasoning Stages.** To assess the contribution of each core component in AGENT KB, we conduct systematic ablation studies in Table 3. Details of the ablation elements can be found in Appendix C.

Table 3: Ablation study for components of the AGENT KB.

| Ablation Setting | Avg | Level 1 | Level 2 | Level 3 |
|---|---|---|---|---|
| smolagent | 55.15 | 67.92 | 53.49 | 34.62 |
| smolagents +AGENT KB | 61.21 | 79.25 | 58.14 | 34.62 |
| w/o Planning Step | 59.39 | 75.47 | 56.98 | 34.62 |
| w/o Feedback Step | 59.39 | 73.58 | 58.14 | 34.62 |
| w/o **Reason** Module | 60.00 | 77.36 | 56.98 | 34.62 |
| w/o **Retrieve** Module | 57.58 | 73.58 | 54.65 | 34.62 |
| w/o **Refine** Module | 55.15 | 69.81 | 53.49 | 30.77 |
| w/ Raw Workflow | 58.18 | 73.58 | 55.81 | 34.62 |

Removing the **Refine** stage incurs the largest drop (−6.06), confirming that retrieved workflows must be adapted rather than replayed. Removing either **Retrieve** pass caps average pass@1 at 59.39%, with sharper Level 1 erosion without the feedback stage (79.25%→73.58%) and without the planning stage (79.25%→75.47%), underscoring their complementary planning/feedback roles. The **Refine** stage imposes the largest penalty when ablated (61.21%→55.15% overall; Level 3 34.62%→30.77%), while dropping **Retrieve** loses 3.63 and **Reason** only 1.21, indicating that knowledge grounding and structured hypothesis drafting together prevent regression even when raw workflow logs (58.18%) are available. Figure 5a further shows AGENT KB benefit from a disagreement gate in the feedback stage with a threshold at $\beta \approx 0.8$.

**Hybrid retrieval outperforms individual similarity metrics.** We compare three retrieval methods: lexical (BM25), semantic (embedding), and hybrid. The hybrid strategy consistently achieves the highest accuracy across general reasoning and software engineering benchmarks, combining the precision of exact matches with the broader coverage of semantic similarity. This complementary fusion proves essential for cross-framework knowledge transfer, as different agent architectures may require either precise tool matches or conceptual similarity depending on the task context.

Table 4: Performance of smolagents (GPT-4.1) on GAIA and SWE-bench with different knowledge types. Baseline uses no augmentation, HAND CRAFTED uses student-annotated experiences, and AGENT KB uses automatically extracted and refined experiences.

| Knowledge type | GAIA | | | | SWE-bench |
| | Average | Level 1 | Level 2 | Level 3 | Lite |
|---|---|---|---|---|---|
| Baseline | 55.15 | 67.92 | 53.49 | 34.62 | 24.33 |
| + HAND CRAFTED | **76.97** | **84.91** | **79.07** | 53.85 | **55.67** |
| + AGENT KB | 75.15 | **84.91** | 74.42 | **57.69** | 51.00 |

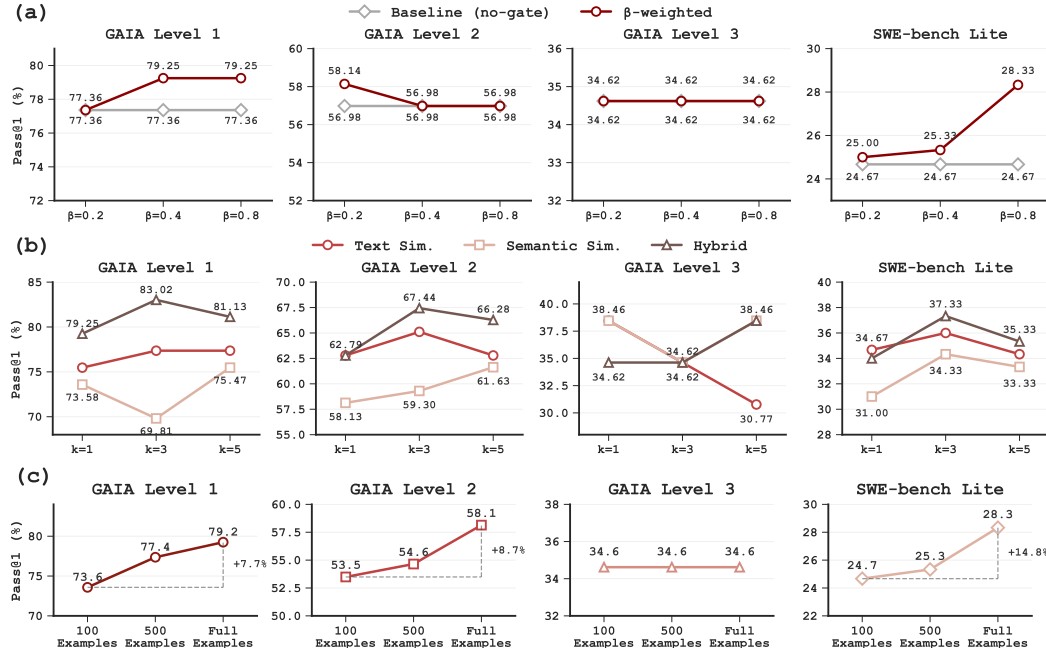

Figure 5: Ablation analysis of retrieval configuration, knowledge-base size, and feedback weighting. (a) Impact of confidence weighting hyper-parameter $\beta$ on feedback integration. (b) Comparison of retrieval strategies across top-$k$ settings. Text similarity, semantic similarity, and hybrid methods are evaluated on GAIA Levels 1–3 and SWE-bench Lite. (c) Effect of knowledge-base size on validation performance.

Figure 5b shows that hybrid retrieval maintains robust performance across different top-$k$ settings, with optimal results at $k = 3$ where it attains peak accuracy on general reasoning tasks (83.0% on GAIA Level 1) while remaining effective across software engineering benchmarks.

**Increasing knowledge base size improves validation performance.** Figure 5c shows performance degrades gracefully as AGENT KB shrinks. With 100 examples, general reasoning and software benchmarks retain capability, indicating that small stores offer useful prior trajectories. Expanding to 500 examples yields consistent gains in reasoning, while software tasks benefit greatly, highlighting the importance of this scale for code repair. Advanced reasoning tasks remain flat, suggest-

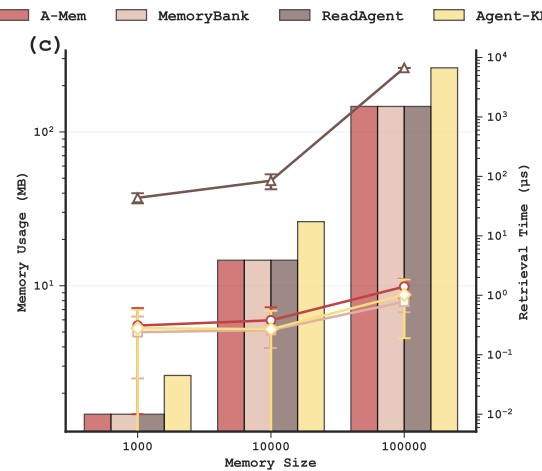

Figure 4: Retrieval latency & memory footprint when scaling different stores. Baselines are taken directly from Xu et al. (2025).

ing that the quality of abstraction is bottlenecked, not the quantity. Larger knowledge bases reliably improve performance, but supporting complex tasks needs better structuring and retrieval.

**Automatic experience construction matches manual curation.** Table 4 shows that AGENT KB's automatically refined knowledge matches hand-crafted experiences (annotated by five computer science students) on GAIA (75.15% vs. 76.97%), surpasses them on Level 3 (57.69% vs. 53.85%), and lifts SWE-bench lite accuracy from 24.33% to 51.00%.

**Latency and memory overhead remain modest.** Figure 4 compares raw lookup latency against store size for AGENT KB and alternatives such as A-Mem (Xu et al., 2025), MemoryBank (Zhong et al., 2024), and

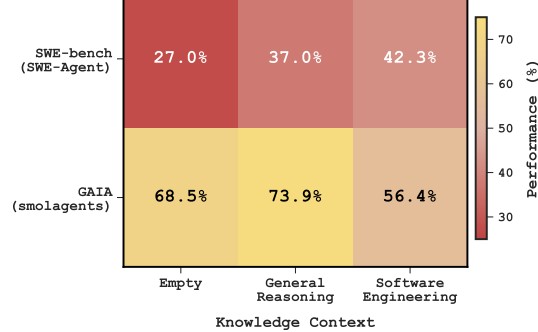

Figure 6: Cross-domain knowledge transfer analysis. Performance comparison when applying domain-specific knowledge bases to different task types.

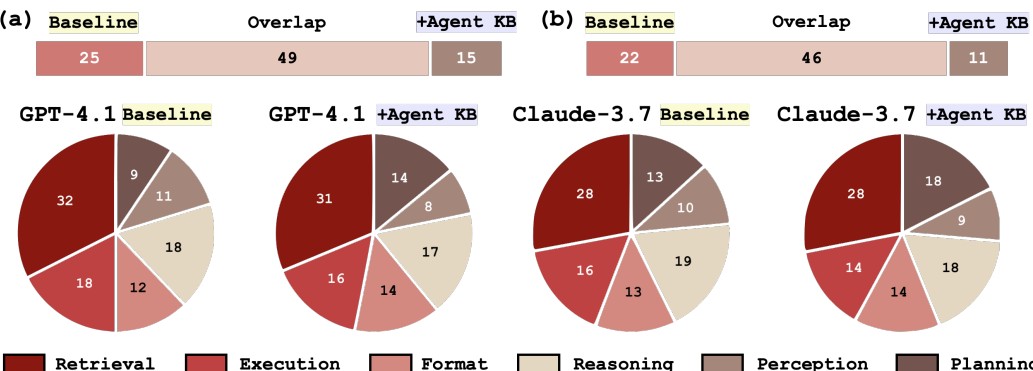

Figure 7: The frequency of errors comparing smolagents (Roucher et al., 2025) with and without AGENT KB on GAIA. The Venn diagrams quantify overlapping and unique failure cases, while the horizontal bar charts show error counts per category. (a) Results for GPT-4.1. (b) Results for Claude-3.7.

ReadAgent (Lee et al., 2024). AGENT KB maintains competitive latency while occupying a similar memory footprint across different store sizes. Alongside the cost accounting in Appendix D, this indicates that AGENT KB's construction and inference overhead remains minor relative to the performance gains.

**Domain-specific knowledge bases exhibit asymmetric transferability.** Figure 6 compares *general reasoning experiences* (e.g., BrowseComp, HopRAG, HLE, and WebWalkerQA) with *software engineering experiences* (e.g., RepoClassBench, SWE-Gym-Raw, and RepoEval). Reasoning experience reaches 73.9% in GAIA and still reaches 37.0% in SWE-bench, whereas SWE experience reaches 42.3% in SWE-bench, but drops to 56.4% in GAIA. This asymmetry shows that SWE knowledge does not generalize to reasoning tasks, while reasoning experience retains partial utility in SWE domains.

## 4.4 ERROR ANALYSIS

On the GAIA benchmark, we analyze error distributions under baseline and AGENT KB-augmented configurations (Figure 7). For GPT-4.1 (Figure 7a), 49 errors are shared across both settings, while 25 are unique to the baseline; AGENT KB introduces only 15 new errors, yielding a net reduction of 10. For Claude-3.7 (Figure 7b), 46 errors persist in both runs, with 22 baseline-specific errors corrected and 11 new errors added, giving a net improvement of 11. We manually categorized each error into six classes: *retrieval* (incorrect or missing evidence), *planning* (invalid task decomposition or step ordering), *reasoning* (logical inconsistency or unsupported inference), *format* (violations of required output schema), *perception* (failures in image/video understanding or tool grounding), and *execution* (extraneous or fabricated steps). Pie charts show the relative prevalence of each type. With GPT-4.1, retrieval errors decrease from 24 to 20 and planning errors from 13 to 10, reflecting more consistent query formulation and workflow reuse. Claude-3.7 achieves larger gains in reasoning, dropping from 13 to 8, alongside fewer retrieval failures (19 to 16). These improvements arise from AGENT KB's knowledge base, which encodes search protocols, planning templates, and formatting conventions, enabling agents to adopt proven strategies rather than improvising from scratch. While perception errors remain constrained by tool capabilities, AGENT KB mitigates their impact by reducing unnecessary steps and minimizing context length. Overall, both models benefit from similar error reductions, with Claude-3.7 excelling in reasoning robustness and GPT-4.1 in perception alignment, underscoring how AGENT KB complements different model strengths on GAIA.

## 5 CONCLUSION

We presented AGENT KB, a cross-framework memory layer that abstracts heterogeneous agent traces into reusable experiences. By coupling *hybrid retrieval* with a *disagreement-gated refinement stage*, it addresses the core challenges of representation heterogeneity, context mismatch, and knowledge interference. Experiments across GAIA, HLE, GPQA, and SWE-bench confirm consistent improvements, with automatically generated experiences performing comparably to curated ones and surpassing them on harder tasks. These results suggest that a *shared, evolving memory backbone* offers a practical step toward collective agent intelligence, with future work aimed at richer modalities and longer-horizon reasoning.

ETHICS STATEMENT

Our work builds on publicly available datasets (GAIA, HLE, GPQA, SWE-bench) and follows their respective licenses. We do not foresee direct ethical concerns; however, when deploying agent memory systems in practice, one should carefully consider data privacy, potential bias in retrieved knowledge, and the risk of misuse in high-stakes domains.

REPRODUCIBILITY STATEMENT

We provide complete details of architectures, configurations, datasets, and evaluation protocols in the main text and the Appendix. Our code and scripts to reproduce all experiments are available at https://anonymous.4open.science/r/Agent-KB/.

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

# Part I

# Appendix

## Table of Contents

# A    EXPERIENCE SOURCE OVERVIEW

Our AGENT KB is constructed from a diverse set of benchmark datasets spanning code reasoning, web navigation, multi-hop retrieval, and human-level evaluation tasks. Each dataset contributes structured experience entries that reflect distinct problem-solving patterns and domain characteristics.

Table 5 summarizes the data sources, their original task counts, and the number of resulting experience entries after processing:

Table 5: Overview of datasets used to construct the experience knowledge base.

| Dataset | Domain | Tasks | Generated Experiences |
|---|---|---|---|
| *General QA experiences* | | | |
| BrowseComp (Wei et al., 2025a) | Web navigation | 1,266 | 1,266 |
| MultiHopRAG (Liu et al., 2025d) | Multi-hop reasoning | 2,556 | 2,556 |
| HLE (Phan et al., 2025) | Expert-level QA | 3,000 | ~2,000 |
| WebWalkerQA (Wu et al., 2025a) | Open-domain QA | 680 | 680 |
| *Software engineering experiences* | | | |
| RepoClassBench (Deshpande et al., 2024) | Code understanding | 100 | 1,000 |
| SWE-Gym-Raw (Pan et al., 2024) | Code generation | 100 | 1,000 |
| RepoEval (Zhang et al., 2023) | Code completion | 100 | 1,000 |
| **Total (approx.)** | | 7,802 | ~9,502 |

**BrowseComp.** We processed all 1,266 tasks from the BrowseComp benchmark (`https://huggingface.co/datasets/smolagents/browse_comp`), creating one experience entry per task. These experiences capture web browsing, information retrieval, and multimodal reasoning patterns.

**MultiHopRAG.** We incorporated all 2,556 tasks from the MultiHopRAG dataset (`https://github.com/yixuantt/MultiHop-RAG/tree/main/dataset`), with each task contributing one experience entry. MultiHopRAG experiences focus on multi-hop reasoning and retrieval-augmented generation scenarios.

**HLE.** From the HLE benchmark's 3,000 tasks (`https://huggingface.co/datasets/cais/hle`), we selected the text-based subset, creating one experience entry per task. We excluded non-textual tasks to maintain consistency in knowledge representation. These experiences cover human-level evaluation scenarios across diverse domains.

**WebWalkerQA.** We integrated 680 tasks from WebWalkerQA (`https://huggingface.co/datasets/callanwu/WebWalkerQA`), with each task contributing one experience entry. These experiences capture web navigation and question-answering patterns in open-domain contexts.

**RepoClassBench.** We utilized the RepoClassBench dataset (`https://github.com/microsoft/repoclassbench`), selecting 100 representative cases from Python repositories that align with those in SWE-bench. For each case, we generated 10 distinct experiences capturing different solution approaches, resulting in 1,000 structured knowledge entries. These experiences focus on repository classification and code understanding tasks.

**SWE-Gym-Raw.** We incorporated the SWE-Gym-Raw dataset (`https://huggingface.co/datasets/SWE-Gym/SWE-Gym-Raw`), from which we selected 100 diverse problem instances. Following a methodology similar to RepoClassBench, we generated 10 distinct experiences per instance, resulting in a total of 1,000 knowledge entries. These experiences primarily focus on code generation and bug-fixing scenarios within Python-based repositories.

**RepoEval.** From the RepoEval dataset (`https://github.com/microsoft/CodeT/tree/main/RepoCoder/datasets`), we selected 100 cases and generated 10 experiences per case, creating an additional 1,000 knowledge entries. RepoEval experiences focus on code completion and repository-level programming tasks in Python.

## B   HAND-CRAFTED EXPERIENCE PROCESS

To identify common failure modes and improve generalization, human annotators manually inspect a subset of failed logs. They summarize recurring issues such as *incorrect tool selection*, *misaligned reasoning chains*, or *missing preconditions or constraints*. These failures are abstracted into correction templates that serve as few-shot examples for the experience generation model. The abstraction process relies on a set of reasoning templates:

---

**AGENT KB data template**

```
{
"question": "<question from various data source>",
"agent_plan": "<Agent original plan>",
"agent_experience": "<detailed agent experience>",
}
```

---

The procedure of hand-crafted experience is described as follows:

- **Step 1: Team Setup and Objective Definition**

  Three computer science students familiar with the GAIA benchmark and agent reasoning workflows were recruited to collaboratively design high-quality prompts. The main objective was to transform successful agent reasoning paths into structured, human-readable instructions that captured essential steps, tools, and decision rules.

- **Step 2: Review of Historical Logs**

  Each student was assigned a subset of GAIA benchmark tasks (Level 1, 2, 3). They thoroughly examined the corresponding smolagent logs, focusing on:

  - Tasks where the agent reached the correct answer.
  - Action sequences that were logically sound and tool-use efficient.
  - Common patterns across multiple tasks.

  After that, they also analyzed the logs of the failed questions, trying to fix the wrong answers by hand with the successful experience.

- **Step 3: Prompt Authoring and Standardization**

  The team synthesized these findings into general reasoning workflows—abstract sequences that could be reused.

  Each reasoning pattern was rewritten into a natural language instructional prompt. Prompts were standardized to use consistent sentence structures, imperative voice, and tool-neutral references.

- **Hand Crafted Example Experience:**

  ```
  Search for the 2015 paper "Pie Menus or Linear Menus, Which Is Better?"
  on a scholarly database (e.g., Google Scholar or IEEE Xplore) and
  note the authors in "First M. Last" format. For each author, look
  up their publication history on DBLP or Google Scholar and list all
  their papers with publication years. Determine which author has works
  published before 2015, and collect that author's prior publications.
  Sort the author's earlier papers by year and identify the very first
  one. Verify the title of that earliest paper against the database
  entry to ensure accuracy.
  ```

- **Step 4: Effectiveness Testing and Selection**

  To evaluate quality, each handcrafted experience was tested via few-shot prompting on similar GAIA tasks.

  The top 80 prompts with the best performance were selected as the canonical set.

- **Step 5: Generalization to Other Benchmarks**

  Using these 80 high-quality examples, we applied few-shot prompting to generate experience instructions for other reasoning benchmarks.

## C    ABLATION DETAILS OF **Reason-Retrieve-Refine** MODULES

To evaluate the effectiveness of each component in our AGENT KB framework, we conduct a series of ablation studies. The deployment loop operates in two retrieval phases with distinct objectives:

- **Planning stage** forms the initial plan. It follows the **Reason-Retrieve-Refine** cycle to summarize the query, select experiences, and weave them into an executable workflow.

- **Feedback stage** reuses the same cycle on execution traces. It reasons over the critic's highlights, retrieves precedents, and refines the plan while guarded by the disagreement gate.

The experimental setup involves systematically removing or disabling specific modules or agents to assess their contributions. The results are summarized in Table 3, with the following definitions:

- w/o Planning Stage: The first-stage steps are removed.

- w/o Feedback Stage: The second-stage steps are removed.

- w/o **Reason** Module: In both stages, no reasoning is performed; only retrieval based on raw data is conducted.

- w/o **Retrieve** Module: Both stages omit the retrieval process entirely. Agents rely solely on prompt-based instructions to generate responses, without consulting prior experiences.

- w/o **Refine** Module: No refinement is performed of both stags; only the retrieved content is used as knowledge.

- w/ Raw Workflow : The full retrieve pipeline is used, but without any explicit modular control— i.e., the model follows a standard prompting strategy throughout, lacking structured guidance through the **Reason** and **Refine** phases.

These ablation experiments provide insight into how each module contributes to overall performance, particularly in terms of accuracy, robustness, and coherence in complex reasoning tasks.

## D    INFERENCE COST BREAKDOWN

Tables 6 and 7 report the token and monetary budget of AGENT KB across GAIA and SWE-bench lite. In GAIA, the retrieval loop adds only $0.27 on top of a full evaluation of $86.0 USD - less than 0.4% of the cost per run. The offline ingestion step is a one-time expense amortized over future runs. On SWE-bench lite, hinting with AGENT KB costs on average <0.004 USD per issue with short prompts (< 7,000 tokens), keeping the marginal overhead well below one cent when paired with GPT-4.1. These results highlight that AGENT KB provides cross-framework experience sharing at negligible additional cost.

Table 6: Per-task token and cost budget for GAIA validation (165 tasks). Shares cover the per-evaluation budget and exclude the one-off knowledge-base ingestion step. Pricing: $1.36 per million prompt tokens and $5.44 per million completion tokens. Values are averaged across 165 tasks.

| Module | Prompt tokens | Completion tokens | Cost (USD) | Share (%) |
|---|---|---|---|---|
| *Evaluation (per GAIA task)* | | | | |
| Action loop | 205,873 | 41,912 | 0.51 | 98.2 |
| Log summary | 6,182 | 67 | 0.01 | 1.6 |
| Planning | 237 | 119 | 0.001 | 0.2 |
| Feedback | 294 | 123 | 0.001 | 0.2 |
| **Total (evaluation)** | **212,586** | **42,221** | **0.52** | **100.0** |
| *One-off setup* | | | | |
| AGENT KB construction | 5,140,655 | 768,270 | 10.88 | – |

Table 7: Per-instance cost for SWE-bench lite when AGENT KB supplies hints (GPT-4.1). Max steps per issue fixed at 100. Token counts are per issue. Token pricing follows the same schedule as Table 6.

| Hint source | Prompt tokens | Completion tokens | Cost (USD) | Hint length |
|---|---|---|---|---|
| RepoClassBench | 6,543 | 912 | 0.0078 | 90 |
| RepoClassBench (refine) | 4,217 | 508 | 0.0028 | 130 |
| Top-$n$ SWE-Gym | 2,847 | 296 | 0.0019 | 60 |
| Top-$n$ RepoClassBench | 3,129 | 402 | 0.0021 | 70 |
| **Average** | **4,184** | **530** | **0.0037** | **88** |

# E    EXAMPLES

This section provides concrete examples and demonstrations of how AGENT KB processes different types of queries and workflows. We present detailed execution examples and comprehensive illustrations of the system's capabilities across various domains and task types.

## E.1    EXECUTION EXAMPLE

Initially, the AGENT KB independently processed a query, leveraging its internal retrieval mechanisms to summarize relevant information. This initial processing led to the generation of a preliminary plan. As part of this plan, the AGENT KB initiated a search for the two most commonly associated chemicals. However, the search successfully retrieved the European Community (EC) number for only one of these chemicals, as the other was not an enzyme. Based on this incomplete retrieval, the AGENT KB incorrectly concluded that the true answer was solely confined to entry "3.1.3.1," leading to two sequential and erroneous returns.

Subsequently, the feedback stage provided a more explicit and refined planning directive, offering a clearer understanding of the task trajectory and expected outcome with fewer steps. It clarified that the question pertained explicitly to enzymes, since only enzymes have EC numbers. As a result, the AGENT KB would no longer search for non-enzyme chemicals in the first place, and it would avoid mistakenly returning a "no correct answer" response in the following steps. This guidance helped the AGENT KB overcome the limitations of its initial independent processing and enabled it to retain and utilize the corrected context.

---

**AGENT KB - User Question**

```
What are the EC numbers of the two most commonly used chemicals for the virus
testing method in the paper about SPFMV and SPCSV in the Pearl Of Africa from
2016? Return the semicolon-separated numbers in the order of the alphabetized
chemicals.
```

---

**AGENT KB - Reason the question to get suggestion**

```
EC numbers; virus testing method; SPFMV; SPCSV; Pearl Of Africa; 2016 paper;
commonly used chemicals; chemical alphabetization
```

---

**AGENT KB - Retrieve from AgentKB**

```
Similar task: During studies of the phage genome database, the researchers
identified an operon......
Suggestions: Explicitly map experimental design to logical variables—use data
frames or grid tables for clarity......
```

**AGENT KB - Initial Plan after refinement**

```
1. Identify the specific 2016 paper on SPFMV and SPCSV in Uganda.
2. Retrieve the full text using inspect_file_as_text or search agent.
3. Check the methodology section for the virus testing method (e.g., ELISA).
4. Extract chemicals used, focusing on enzymes.
5. Determine the two most commonly used ones.
6. Alphabetize the chemical names.
7. Look up their EC numbers using BRENDA, ExplorEnz, etc.
8. Verify EC number correctness.
9. Return EC numbers in the correct order, separated by a semicolon.
```

**AGENT KB - Model Prediction**

```
3.1.3.1;3.1.3.1
```

**AGENT KB - Retrieve from AgentKB**

```
Similar task: Which of the following peptides has been found to bind or is
highly likely to bind an integrin receptor in an in vitro assay......
Suggestions: Always decompose the question to clarify the biological context
and functional motifs encountered in sequence-based tasks......
```

**AGENT KB - feedback stage's suggestion**

```
1.  Search for the paper using keywords like "SPFMV", "SPCSV", "Pearl of
Africa", and "2016".
2. Identify the virus detection method (typically ELISA) and the enzymes used
in the labeling step......
```

**AGENT KB - New Plan**

```
1. Use compact keywords to Search for the 2016 paper on SPFMV and SPCSV in
Pearl of Africa.
2. Review the method section to find the virus detection technique and the
two enzymes.
3. Look up EC numbers from BRENDA or ExPASy.
4. Alphabetize enzyme names.
5. Return EC numbers joined by a semicolon.
```

**AGENT KB - Model Thought**

```
I suspect the two most common enzymes for virus detection are alkaline
phosphatase and horseradish peroxidase. Their EC numbers are:
* Alkaline phosphatase: 3.1.3.1
* Horseradish peroxidase: 1.11.1.7
Alphabetically, alkaline phosphatase comes first.
So I will return: 3.1.3.1;1.11.1.7
```

**AGENT KB - Final Output (Code)**

```
answer = "3.1.3.1;1.11.1.7"
final_answer(answer)
```

**AGENT KB - Observation**

```
Last output from code snippet:
3.1.3.1;1.11.1.7
```

## E.2 COMPREHENSIVE EXAMPLES

This section provides concrete examples of how AGENT KB processes and stores different types of agent experiences. We demonstrate three key components: SWE-bench workflow examples showing problem-solution pairs, raw execution logs transformed into structured experiences, and complex multi-constraint query processing.

### E.2.1 AN EXAMPLE ON SWE-BENCH

The following examples illustrate how AGENT KB stores and retrieves domain-specific knowledge for software engineering tasks from the SWE-bench dataset. Each example shows a problem description paired with relevant guidance retrieved from AGENT KB, demonstrating the system's ability to provide contextual assistance for code debugging and modification tasks.

---

**AGENT KB - Problem Description**

```
In the project that automatically generates API documentation for Python
projects, an extra backslash is inserted before underscores—for example, hello_
is rendered as hello\_
```

---

**AGENT KB - Retrieve from AgentKB**

```
Check the string processing part in the relevant functions to ensure that
escape is only carried out when necessary. For example, whether all parameters
ending with _ need to be escaped, or whether different handling methods are
required in certain specific contexts (such as attribute names, parameter
names, etc.).
When modifying conditions, not only the original conditions should be taken
into account, but also factors such as configuration and context should be
combined to ensure the accuracy of the logic.
```

---

**AGENT KB - Problem Description**

```
Disabling evaluation globally with with evaluate(False) interferes with
sympify's string-parsing logic, preventing some integer expressions from being
instantiated as integer objects.
```

---

**AGENT KB - Retrieve from AgentKB**

```
When adding or modifying a conditional check (such as for 'evaluate' or
imaginary coordinates), ensure the logic does not inadvertently skip important
validation for invalid inputs (such as actual imaginary numbers), and only
disables overly strict checks for valid real inputs. This is critical to
maintain mathematical correctness while fixing the bug. (Most important)
When changing the logic in constructors (like Point/Point2D), verify that the
minimal change solves the immediate bug, does not introduce new regressions,
and does not allow forbidden cases (e.g., actual imaginary coordinates)
```

---

These examples demonstrate AGENT KB's ability to provide targeted guidance for common software engineering challenges. The first example addresses API documentation generation issues related to string escaping, while the second focuses on debugging a symbolic mathematics library. Notice how the retrieved knowledge provides specific, actionable advice rather than generic troubleshooting steps.

### E.2.2 RAW LOG TO EXPERIENCE GENERATION

This subsection demonstrates the complete pipeline for transforming raw agent execution logs into structured knowledge that can be stored in AGENT KB. This process is crucial for the system's learning capability, allowing successful problem-solving strategies to be captured and reused.

**Raw Log Example**    The following demonstrates how agent execution logs are processed and transformed into structured experiences for AGENT KB. This particular example shows a bioinformatics task involving protein structure analysis, where the agent had to adapt its approach when encountering unexpected file formats.

```
1  {
2    "agent_name": "gpt-4.1",
3    "question": "Using the Biopython library in Python, parse the PDB file of the
       protein identified by the PDB ID 5wb7 from the RCSB Protein Data Bank. Calculate
       the distance between the first and second atoms as they are listed in the PDB
       file. Report the answer in Angstroms, rounded to the nearest picometer.",
4    "prediction": "1.46",
5    "true_answer": "1.456",
6    "intermediate_steps": [
7      {
8        "task": "You have one question to answer...",
9        "step_type": "task"
10     },
11     {
12       "facts": "Here are the facts that I know so far...",
13       "plan": "Here is the plan of action that I will follow...",
14       "step_type": "planning"
15     },
16     {
17       "tool_calls": [{"id": "call_1", "type": "function", "function": {"name":
         "python_interpreter", "arguments": "..."}}],
18       "error": {"type": "AgentExecutionError", "message": "Code execution failed..."},
19       "step_type": "action"
20     }
21   ]
22 }
```

Listing 1: Raw Agent Execution Log

**Key Insights**    These examples collectively demonstrate several important aspects of AGENT KB's design and functionality:

1. **Domain Adaptation:** The system successfully captures domain-specific knowledge across different fields (software engineering, bioinformatics, biographical research), showing its general applicability.

2. **Error Recovery:** Raw logs show how agents adapt when initial approaches fail, and these adaptation strategies are preserved as valuable experiences for future use.

3. **Precision Management:** The system learns specific formatting and precision requirements, crucial for tasks requiring exact numerical outputs.

4. **Multi-Agent Coordination:** Complex queries demonstrate how different agent roles (general reasoning vs. specialized search) can be coordinated with distinct but complementary planning strategies.

5. **Source Validation:** The emphasis on authoritative sources and cross-validation shows the system's commitment to reliability and accuracy in information gathering.

These examples illustrate how AGENT KB transforms individual agent experiences into a shared knowledge resource that enhances the performance of the entire agent ecosystem.

From the raw log, the following agent experience is extracted:

> **Agent Experience**
>
> When calculating distance, extract only the first two ATOM coordinates directly without skipping any lines. If the task asks for the distance in Ångströms, rounded to the nearest picometer, keep the original Å value with three decimal places precision, without converting back and forth between Å and pm. Output the

distance directly in Ångströms, keeping the computed value with three decimals (e.g., x.xxx), do not round it to fewer decimal places. The output is just the number without any units or symbols, such as x.xxx, not x.xxx Ångströms.

### E.2.3   COMPLEX QUERY PROCESSING EXAMPLE

**Complex Query Example**

**Question:** There's this popular figure with multiple Grammy awards in the entertainment industry, who put out their first album before 1969, had substance dependence, and was dismissed from school before they turned 20. Their first life partner died in 1997. They became uniformed personnel at some point in their lives. What's the name of the hospital they died?
**True Answer:** St. John's Health Center

**Generated JSON Structure**   The following JSON structure shows how AGENT KB organizes planning strategies and experiences for complex queries. The structure includes separate planning approaches for different agent types (general agents vs. specialized search agents) and captures both successful strategies and lessons learned from the task execution. This structured approach enables systematic knowledge transfer and strategy refinement.

```json
{
  "question": "There's this popular figure with multiple Grammy awards...",
  "true_answer": "St. John's Health Center",
  "agent_planning": "1. Parse the question to extract all key constraints: multiple
    Grammy awards, first album before 1969, substance dependence, dismissed from
    school before age 20, first life partner died in 1997, served as uniformed
    personnel, determine place/hospital of death.\n2. Conceptual plan:\n- Identify
    the possible entertainers matching all constraints.\n- For each candidate:\n  a)
    Verify the timeline for first album release (before 1969)\n  b) Check Grammy
    history\n  c) Search biographical records for substance abuse and educational
    background\n  d) Confirm information about life partner's death year and
    uniformed service\n  e) Pinpoint the date and location/hospital of death of the
    matched figure.",
  "search_agent_planning": "1. Receive precise person identifier from Code Agent or
    use biographical clues to triangulate the subject.\n2. Formulate search queries
    for identification and specific hospital information.\n3. Prioritize official
    biographical sources, reputable news outlets, Grammy records.\n4. Cross-check
    critical data points to validate subject match.\n5. Extract facts about location
    and hospital of death from obituaries.",
  "agent_experience": [
    "Break down multifaceted questions into smaller constraint checks",
    "Explicitly log and verify biographical constraints with multi-source
    confirmation",
    "Select high-reliability sources for biographical and award data",
    "Delegate to Search Agent early with specific sub-queries",
    "Validate final answers by chaining all found facts back to original constraints"
  ],
  "search_agent_experience": [
    "Decompose complex queries into sequential search refinements",
    "Craft highly specific queries for ambiguous identifiers",
    "Favor authoritative sources over entertainment/tabloid content",
    "Cross-validate information from multiple independent sources",
    "Format results with direct attribution and clear source references"
  ]
}
```

Listing 2: Generated Agent Planning and Experience JSON

# F  COLLECTIONS OF USED PROMPTS

## F.1  PROMPT DESIGN FOR AGENT KB CONSTRUCTION

### F.1.1  GENERAL TASKS

---

**AGENT KB Generation Prompt**

You will act as an advanced AI evaluation system tasked with analyzing a complex problem that an agent handles. Your analysis will extract valuable insights from this process. Follow these instructions carefully:

1. I will provide a question and its correct answer (true_answer).

2. First, simulate the agent's planning process in detail. Describe how it would:
- Break down the problem into logical components
- Determine which tools to use (code execution, data processing, API calls)
- Decide when to delegate to the Search Agent
- Plan data transformations and analysis steps
- Structure the final solution
Include specific reasoning steps, potential code snippets considered, and decision points. Only include content in the agent plan, without any other description.

4. Next, based on the question and your simulated planning processes, create a realistic error scenario. Describe:
- Where and how the agents might fail
- Incorrect assumptions they might make
- Data misinterpretations or code errors
- Logical flaws in their approach

5. Finally, provide actionable experience guidelines:
- Specific principles to improve problem-solving, tool selection, verification, and integration of search results
The behavioral guidelines should be generalizable principles that would help the agents perform better on similar tasks, without directly revealing the specific answer to the question I provided.

Output your complete analysis in the following JSON format with no additional text:
{
  "question": "<question I provide>",
  "true_answer": "<correct answer I provide>",
  "agent_plan": "<your detailed Code Agent plan simulation>",
  "agent_experience": "<your actionable Code Agent guidelines>",
}

Here is an example:

{
  "question": "<question from hand-crafted experience pool>",
  "true_answer": "<correct answer>",
  "agent_plan": "<Real Code Agent plan>",
  "agent_experience": "<Hand-crafted agent experience>",
}

---

## F.1.2  GAIA

---

**AGENT KB Generation Prompt**

You will act as an advanced AI evaluation system tasked with analyzing a complex problem handled by a Code Agent with an embedded Search Agent. Your analysis will extract valuable insights from this process.  Follow these instructions carefully:

1. I will provide a question and its correct answer (true_answer).

2. First, simulate the Code Agent's planning process in detail. Describe how it would:
- Break down the problem into logical components
- Determine which tools to use (code execution, data processing, API calls)
- Decide when to delegate to the Search Agent
- Plan data transformations and analysis steps
- Structure the final solution
Include specific reasoning steps, potential code snippets considered, and decision points.  Only include content in the agent plan, without any other description.

3. Next, simulate the Search Agent's planning process in detail. Describe how it would:
- Parse the search query requirements from the Code Agent
- Formulate effective search queries
- Determine which sources to prioritize
- Extract and validate relevant information
- Process and structure the search results for the Code Agent
Include specific query formulation strategies and information filtering approaches. Only include content to search the agent plan, without any other description.

4.  Based on the question and your simulated planning processes, create a realistic error scenario. Describe:
- Where and how the agents might fail
- Incorrect assumptions they might make
- Data misinterpretations or code errors
- Logical flaws in their approach

5. Finally, provide two sets of actionable experience guidelines:
- For the Code Agent: Specific principles to improve problem-solving, tool selection, verification, and integration of search results
- For the Search Agent: Specific principles to enhance query formulation, source evaluation, information extraction, and result formatting
The behavioral guidelines should be generalizable principles that would help the agents perform better on similar tasks, without directly revealing the specific answer to the question I provided.

Important: If the question does not require the search agent to solve, leave "search_agent_plan" and "search_agent_experience" empty in your response.

Output your complete analysis in the following JSON format with no additional text:
{
"question": "<question I provide>",
"true_answer": "<correct answer I provide>",
"agent_plan": "<your detailed Code Agent plan simulation>",

---

```
"search_agent_plan": "<your detailed Search Agent plan simulation>",
"agent_experience": "<your actionable Code Agent guidelines>",
"search_agent_experience": "<your actionable Search Agent guidelines>"
}

Here is an example:

{
"question": "<question from hand-crafted experience pool>",
"true_answer": "<correct answer>",
"agent_plan": "<Real Code Agent plan>",
"search_agent_plan": "<Real Search Agent plan>",
"agent_experience": "<Hand-crafted agent experience>",
"search_agent_experience": "<Hand-crafted search agent experience>"
}
```

### F.1.3 SWE-BENCH

**AGENT KB Generation Prompt**

You are an advanced code repair analysis system tasked with constructing structured experiences for Agent KB from SWE-bench tasks. Given a natural language problem description, a model-generated fix, and supporting repair hints, follow the steps below to extract reusable knowledge entries. Your output should conform strictly to JSON formatting and follow the key structure outlined in each step.

1. **Code Reconstruction:**
Given a detailed natural language description of a Python class or function, generate its correct implementation. Ensure it is complete and syntactically valid.
Output key: "code"

2. **Error Analysis and Repair Principles:**
You are given two versions of code: one with errors and one corrected. Analyze the differences and identify key problems in the faulty version. Based on this comparison, produce a list of 10 code repair precautions. These should be generalizable principles addressing common issues (e.g., indentation, type conversion, exception handling, logic errors). Avoid using titles; just output the explanations.
Output key: "hints" (as a list of 10 strings)

3. **Hint Classification:**
Each natural language hint is used to prompt the LLM to repair the code. Classify each hint into one repair category (e.g., "syntax", "logic", "exception handling"). Also, extract important keywords and write a one-sentence summary of the hint.
Output keys: "category", "keywords", "summary"

4. **Repair Type Identification:**
Given the original problem description, identify the {K} most relevant categories this code repair case falls under. Select from a pre-defined set of bug types.
Output key: "categories" (as a list of {K} strings)

```
5. Most Relevant Hints Ranking:
You are given a set of all the hints provided to the model. Analyze the model's
generated fix and its reasoning trace. Based on this analysis, identify the
{N} most relevant hints. These may be either positively helpful or misleading.
Sort them in order of influence on the final patch.
Output key: "hints" (as a list of {N} strings)

Important Notes:
- Always respond strictly in JSON format.
- Do not include section titles, markdown formatting, or explanations.
- When code is requested, return only the code inside the JSON key.
- If any step is not applicable (e.g., hint classification not possible), return
  an empty string or array for that field.
```

## F.2 PROMPT DESIGN FOR AGENT KB PIPELINE

### F.2.1 GAIA

**AGENT KB Reason Prompt**

```
Analyze similar tasks and past experiences to generate concise, actionable
suggestions for improving the current plan. Based on the patterns identified
in relevant tasks and insights from the Agent KB, provide specific
recommendations.

Key Requirements:
1. Focus exclusively on technical/behavioral improvements derived from similar
task patterns and experience.
2. Provide root-cause solutions and implementation strategies based on past
successes.
3. Format output strictly as:
{1. Specific suggestion 1}
{2. Specific suggestion 2}
...

No headings, explanations, or markdown.
You can refer to similar tasks, plans, and corresponding experience to provide
your suggestions:
{
"question": "<Question retrieved from Agent KB>",
"agent_plan": "<Retrieved agent plan>",
"agent_experience": "<Retrieved agent experience>",
}
...
```

**AGENT KB Refine Prompt**

```
Analyze the execution logs to determine the causes of the agent's incorrect
responses. Based on the findings of the log and insights from the provided
similar tasks and experience, generate some concise, actionable suggestions
that the agent must follow to improve accuracy.

**Key Requirements:**
1. Focus exclusively on technical/behavioral fixes derived from log patterns
and the Agent KB.
2.  Provide root-cause resolution (e.g., code logic, data validation, API
```

```
handling) as well as generic advice.
3. Format output strictly as:
{1. Specific suggestion 1}
{2. Specific suggestion 2}
...
No headings, explanations, or markdown.
You can refer to similar tasks and corresponding experience to provide your
suggestions:
{
"question": "<Question retrieved from Agent KB>",
"agent_plan": "<Retrieved agent plan>",
"agent_experience": "<Retrieved agent experience>",
}
...

Execution logs summary:
<Log summary>
```

### F.2.2  SWE-BENCH

**AGENT KB Reason Prompt**

```
Extract key information from user queries to construct efficient search terms
for retrieving the most relevant results.
Requirements:
Analyze the user's question to identify core concepts, terminology, and
keywords Extract contextual information and constraints that may impact
search quality Break down complex questions into searchable components

Identify the domain, subject matter, and specific needs of the question
Output format:
{<core concepts or topics of the question>}

Ensure search terms are specific enough to retrieve relevant information while
maintaining sufficient breadth to capture related cases. Combine technical
terminology with everyday expressions to optimize search effectiveness.
```

**AGENT KB Retrieve Prompt**

```
Given the current bug description, initial patch plan, and model thought
process, retrieve the most relevant historical experiences from Agent KB.
Retrieval Priorities:
1. Prefer experiences with similar bug types (e.g., off-by-one errors, null
pointer exceptions, wrong return value).
2. Favor patches with successful unit test outcomes and generalizable fix
patterns.
3. Include agent plans that show tool usage, exception guards, or correct
interface assumptions.

Format each retrieved experience as:
{
"question": "<SWE-bench issue title or commit description>",
"agent_plan": "<Historical high-level patch or thought process>",
"agent_experience": "<Failure modes avoided or debug strategies that worked>"
}
...
```

Retrieve 3 to 5 relevant entries and return them in the above format for use
in downstream reasoning and refinement.

---

**AGENT KB Refine Prompt**

Analyze the execution trace of the model's patch attempt and identify the
reasons for its failure. You are given: a natural language description of
a code fix problem, the model-generated fix, the model's internal thought
process, and the prompts previously provided to guide the model.
Based on this information, identify the most likely cause of the error and
determine which hints or prompt components influenced the model's incorrect
reasoning. Rank the provided prompts in order of their influence over the
model's behavior.
**Key Requirements:**
1.  Focus exclusively on technical root causes, such as incorrect API
assumptions, scope misunderstanding, faulty patch structure, or missing
validation.
2. Identify which prompt(s) led the model astray, based on reasoning steps or
patch behaviors.
3. Output a strictly ranked list of prompts or hints, based on their importance
in shaping the erroneous behavior.
4. Justify the ranking based on model thought content and the specific failure
observed.

Format strictly as:
{
1. "<Most influential prompt or hint snippet>"
2. "<Second most influential prompt or hint snippet>"
...}

Do not include headings, explanations, or markdown. Focus only on returning
the ranked list with brief justifications inline.

## G LANGUAGE MODEL USAGE

This section outlines the specific roles of large language models (LLMs) within our AGENT KB framework and experimental methodology. We provide detailed documentation of all LLM applications for transparency.

### G.1 SYSTEM COMPONENTS

We employ LLMs in three core functions: (1) **Experience synthesis** using LLMs with few-shot prompting to transform heterogeneous agent logs into standardized representations, (2) **Knowledge curation** through LLM-based ranking when deduplicating similar entries ($\tau = 0.8$ threshold), and (3) **Query processing** for both task analysis in the `Reason` phase and experience adaptation in the `Refine` phase.

### G.2 EXPERIMENTAL SETUP

**Agent Backbones.** Our evaluation involves four distinct agent frameworks, each powered by different LLM configurations: smolagents (`GPT-4o`, `GPT-4.1`, `Claude-3.7`, `Qwen 3-32B`, `DeepSeek-R1`), OWL (`GPT-4o`), SWE-Agent (`GPT-4.1`, `o3-mini`), and OpenHands (`GPT-4o`, `o3-mini`, `GPT-4.1`, `Claude-3.7`, `Qwen 3-3B`, `DeepSeek-R1`). The AGENT KB system acts as a model-agnostic memory layer, interfacing through standardized APIs without requiring changes to agent architectures.

**Evaluation Methodology.** Performance assessment relies solely on ground-truth task completion metrics. We use exact match accuracy for GAIA reasoning tasks and test passage rates for SWE-bench code repair, rather than LLM-generated evaluation scores.

### G.3 MANUSCRIPT DEVELOPMENT

In line with conference transparency standards, we disclose that large language models assisted in manuscript preparation through editorial tasks such as grammar correction, typo detection, and prose clarity improvement. All technical contributions, experimental design, results interpretation, and scientific claims are entirely authored by the researchers.

### G.4 METHODOLOGICAL CONSIDERATIONS

**Computational Overhead.** LLM inference incurs measurable costs during experience construction and retrieval. Cost analysis (Appendix D) estimates these overheads at $3.0 − −$4.5 per task, which is acceptable considering the performance gains of 4.0–18.7 percentage points.

**Architectural Independence.** Although individual agent frameworks depend on specific LLMs, AGENT KB maintains an architecture-agnostic design. Knowledge transfer across frameworks happens via semantic embeddings and standardized action vocabularies, ensuring portability across different model families and API interfaces.

