# OpenReview forum: "Agent KB: Leveraging Cross-Domain Experience for Agentic Problem Solving"
_ICLR.cc/2026/Conference — ICLR 2026 Conference Withdrawn Submission_

### Official Review · Reviewer_HcSR · 2025-10-28

**Soundness:** 2
**Presentation:** 3
**Contribution:** 2
**Rating:** 2
**Confidence:** 4

**Summary:**

This paper proposes Agent KB, a cross-framework memory infrastructure that enables knowledge sharing across heterogeneous agent systems. The core contribution is a plug-and-play knowledge base that captures agent experiences and makes them available to multiple frameworks (smolagents, OpenHands, OWL, SWE-Agent) through a Reason-Retrieve-Refine cycle, augmented with a disagreement gate mechanism. The authors demonstrate substantial improvements across GAIA, HLE, GPQA, and SWE-bench benchmarks.

**Strengths:**

1. This paper touches an important and timely research problem, about how to conduct agent memory effectively for complicated agentic tasts.
2. The peformance reporting on GAIA and SWE-bench is promising and encouraging.
3.The system seems plug-and-play, which can be used by different scalffold and show promising results.

**Weaknesses:**

1. The novelty is limited and very enginerring-heavy:
- Reason-Retrieve-Refine is largely borrowed from case-based reasoning literature
- hybrid retrieval (BM25 + semantic) is standard practice and Few-shot experience generation uses straightforward prompting
- They disagreemnt threshold is based on embedding which is very common in many RAG works, and it's not clear to illustrate how to set this threshold, which may tuned based on target data.

2. The experiments are weak:
- There are no comparison with other memory-augmented methods so the performance gain could be from human prompt engineering.
They only compare with vanilla agents. For example, even if authors mentioned [1] which are very similar to this work, they do not compare directly. For instance, [1] also propose methods or pipeline of constructing memory and retrieve, they also evaluate methods on SWE-Bench. Authors didn't compare in the same setting.

3. Some of the statements are not with enough evidence:
- Claims of "zero-shot transferability" are questionable when experiences come from the same benchmark distributions such as HLE used for construction, HLE Bio/Chem for evaluation
- The framework is tested only on language/code tasks; claims of general applicability are not supported, such as not include MCP-based, math, or reasoning-relevant tasks. Learn from [1] for benchmark picking.

[1] Learn-by- interact: A data-centric framework for self-adaptive agents in realistic environments. ICLR 2025

**Questions:**

1. Learn-by-Interact proposes similar memory construction/retrieval and evaluates on SWE-Bench what is main difference compared to that, and what is comparison with that?
2. How about performance on broader agent tasks including MCP-based benchmarks, math, tau-bench, those OOD benchmark?
3. Without ablating multi-pass without retrieval, how much improvement actually comes from Agent KB versus just multiple attempts?

---

### Official Review · Reviewer_DpVY · 2025-10-30

**Soundness:** 3
**Presentation:** 3
**Contribution:** 2
**Rating:** 4
**Confidence:** 3

**Summary:**

The paper introduces Agentt KB, a cross-framework memory infrastructure that allows heterogeneous AI agent systems (e.g., smolagents, OpenHands, OWL, SWE-Agent) to share execution experiences without retraining. It abstracts trajectories from heterogeneous agent frameworks into structured “experience units,” retrieves relevant ones during planning and feedback, and filters them through a disagreement gate to prevent harmful interference.

I have a few doubts that I seek clarification for. I'll adjust my ratings based on the author's response.

**Strengths:**

The research question is well motivated, aiming to target 3 challenges: representation heterogeneity, context mismatch, and knowledge interference. This is an impactful problem.
Strong quantitative empirical results, with consistent improvements compared to prior memory-based systems of A-MEM.

**Weaknesses:**

1. The ability to abstract and distill the "heterogeneous agent trajectories into structured experience units" is a key concept that the pipeline relies on, and it is implemented by "few-shot prompting (10-15 human-curated exemplars per domain)" and "standardized action vocabularies". This is somewhat brittle to claim the method is seamless if left unjustified. The reliance on "standardized action vocabularies" seems to hide a great deal of complexity. Does this mean the system can only integrate frameworks that already share a similar ontology, or does it require a new manual mapping for every framework? And also, how much human effort is really required to onboard a new, unseen agent framework with a completely novel action space?
2. How is the reward signal $r_j$ attributed back to the specific experiences that were retrieved, especially when multiple (top-k=3) are used?
3. The paper claims cross-framework sharing “without retraining,” but some datasets (e.g., GAIA, SWE-Gym, RepoEval) are used both for constructing the KB and for evaluation. Is there a possible contamination issue here? Even if “held-out labels” are excluded, trajectory-level leakage might inflate results.
4. LLM-based ranking and index maintenance also introduce a non-trivial price. It would be helpful to account for these. In addition, only OpenAI pricing is included in the Table.
5. In Fig. 4, the retrieval latency & memory footprint when scaling different stores are shown. How does the retrieval quality change when scaling up the memory store?
6. Would the prevention of knowledge interference via the cosine similarity-based disagreement gate also filter out potentially useful but diverse experiences, like those "out-of-the-box" examples that could help with exploration?

Nit (does not affect rating):

Fig. 1 contains a professional term "ANISOU" in the Protein Databank. Readers of machine learning papers may not be so familiar with what it means. But I definitely agree that the example is superb (to those who understand Protein Databank).
Figure 5 uses too much vspace. And there seems to be a disorder of the Figure sequence, with Figure 5 appearing before Figure 4

Also, cross-framework experience sharing raises data provenance concerns, especially when logs originate from proprietary agents. The author should acknowledge this.

**Questions:**

Same as the weaknesses.

---

### Official Review · Reviewer_ensa · 2025-10-31

**Soundness:** 2
**Presentation:** 2
**Contribution:** 2
**Rating:** 4
**Confidence:** 3

**Summary:**

The paper presents AGENT KB, a universal and cross-framework memory system designed to address knowledge fragmentation among AI agents. Its core innovation is a structured knowledge base that consolidates agent trajectories into standardized, reusable “experience units.” During inference, agents interact with AGENT KB through a two-stage Reason–Retrieve–Refine process. In the Planning stage, the agent retrieves high-level workflows from the knowledge base to form an initial plan. In the Feedback stage, it uses execution traces to obtain targeted refinements and corrections. Experimental results across benchmarks such as GAIA, HLE, GPQA, and SWE-bench show notable performance improvements across multiple agent systems, demonstrating the approach’s plug-and-play generality.

**Strengths:**

1. The motivation—enabling collective intelligence across different agent frameworks—is timely and well-justified.
2. The paper shows consistent and substantial gains across diverse benchmarks, agent types, and model backbones, with convincing ablation studies to support claims.

**Weaknesses:**

1. The framework-agnostic experience representation appears to be the key innovation, yet its implementation details and technical challenges are not clearly explained (after reading the appendices).
2. The disagreement gate for refinement rejection, another key contribution, seems heuristic-driven; it may wrongly reject beneficial refinements (that might make the embedding similarity low) if the initial plan is flawed.
3. While the experiments are extensive, the paper’s readability and structure make it difficult to parse the results and insights.

**Questions:**

1. How is the utility score used for the memory eviction process in the self-evolving memory section computed?
2. Have you tested AGENT KB with different numbers and choices of seed trajectories (e.g., smaller than 80), and how does this affect final performance? What was the rationale behind selecting 80 seeds?

---

### Official Review · Reviewer_zG4p · 2025-11-05

**Soundness:** 2
**Presentation:** 3
**Contribution:** 2
**Rating:** 4
**Confidence:** 3

**Summary:**

The paper introduces Agent KB, which is a new memory system that shares memory across different agent frameworks that doesn't require additional training. This memory is accessible via API. The authors collect trajectories from multiple agent frameworks (smolagents, OWL, SWE-Agent, OpenHands) across several evals. Using Agent KB improves performance on major evals across both heterogenous and similar model families.

**Strengths:**

The authors built a memory system that demonstrates performance improvements on evals across different model families, frameworks, and eval types. Using cross-framework memory store is new and builds on prior memory stores that focus more on a specific framework. By collecting trajectories across multiple frameworks, the overall system can benefit from the diversity coming from different frameworks. Separating planning from feedback is also relatively new. Evaluation across a number of domains is also a strength of the paper. There are also a number of sound ablations to demonstrate robustness of the findings.

**Weaknesses:**

This paper could be substantially stronger if the improvements from Agent KB style memory were compared to gains from other memory systems (e.g., other classic RAG systems / embedding database, scratchpad, or other API-based memory store). This would help establish novelty compared to other similar systems, and make clearer why this approach is superior and worth continuing to push on.
The authors evaluate on SWE-Bench from 2023, which is known to have ~50% broken problems and is not the community consensus version of the benchmark. Using the SWE-Bench Verified gold 500 set would help make clearer if the pass@1 improvements are actually SoTA or whether the numbers are biased by the broken problems.
The tables of results would also benefit from including confidence intervals instead of just absolute accuracy %.

**Questions:**

Why did the authors pursue this particular memory-based system and why is it better than current systems in use?

---

### Note · Authors · 2026-01-05

I have read and agree with the venue's withdrawal policy on behalf of myself and my co-authors.